# Design and Research of a Strain Elastic Element with a Double-Layer Cross Floating Beam for Strain Gauge Wireless Rotating Dynamometers

**DOI:** 10.3390/mi15070857

**Published:** 2024-06-30

**Authors:** Qinan Wang, Wenge Wu, Yongjuan Zhao, Yunping Cheng, Lijuan Liu, Kaiqiang Yan

**Affiliations:** School of Mechanical Engineering, North University of China, Taiyuan 030051, China; b20220202@st.nuc.edu.cn (Q.W.); zyj@nuc.edu.cn (Y.Z.); ypchengbk@163.com (Y.C.); liulijuan@nuc.edu.cn (L.L.); b20230205@st.nuc.edu.cn (K.Y.)

**Keywords:** double-layer cross floating beam, strain elastic element, strain gauge wireless rotary dynamometer, cutting force

## Abstract

Cutting force is one of the most basic signals that can reflect the information of the cutting process, so it is very necessary to study the strain elastic element of strain gauge wireless rotating dynamometers. This paper proposes a strain elastic element with a double-layer cross floating beam that can be applied to the strain gauge wireless rotating dynamometer, which can simultaneously obtain the four-component cutting force/torque information of $F_{X}$, $F_{Y}$, $F_{Z}$, and $M_{Z}$. Based on the proposed strain elastic element, a compact strain gauge wireless rotating dynamometer is designed, which is composed of a tool holder, upper connection flange, strain elastic element, lower connection flange, tool base, and data acquisition and wireless transmission system. The static model of the double-layer cross floating beam on the strain elastic element is established by the segmented rigid body method, and the relationships between the material, force, structural parameters, and the strain and deformation of the floating beam are obtained. The static model is consistent with the finite element solution, which proves the rationality of the static model. Based on the established static model, the sequential quadratic programming algorithm is used to optimize the structural parameters of the double-layer cross floating beam to maximize the sensitivity of the floating beam. The overall structure of the strain elastic element is analyzed by finite element software, and the strain of the structure under simulation conditions is obtained, which provides a reference for subsequent calibration tests and circuit design. The calibration matrix and dynamic performance of the strain elastic element are obtained by the static calibration test, dynamic calibration test, and cutting test. The results show that the proposed strain elastic element has high sensitivity and low cross-sensitivity error, and can be applied to the strain gauge wireless rotating dynamometer to measure medium- and low-speed cutting forces.

## 1. Introduction

In intelligent manufacturing, cutting process monitoring is a very important aspect [1,2]. The cutting force signal is one of the most basic signals that can reflect the cutting process. It can be used for cutting force prediction [3], exploring material removal mechanisms [4], chatter prediction [5], tool status monitoring [6], surface roughness prediction [7], etc.

In order to meet the demand for measuring cutting force during cutting, wireless rotating dynamometers have been favored by researchers [8]. Wireless rotating dynamometers can be divided into piezoelectric [9], PVDF [10,11], strain [12,13], capacitance [14,15], FBG [16], etc., according to different measurement principles. Among them, the strain gauge wireless rotating dynamometer has been widely studied due to its advantages of low cost, stable signal, and high reliability. Rizal [17,18] proposed a strain gauge wireless rotating dynamometer with a three-dimensional cross beam strain elastic element, which realizes the online monitoring of the three-component cutting force. Zhang et al. [19] proposed an elastic element that separates the measuring part (thin beam) from the supporting part (thick beam), which realizes the measurement of three cutting forces and spindle torque during cutting. In summary, the strain elastic element is the core of the strain gauge wireless rotating dynamometer. However, the elastic element used in the above dynamometer has a large axial dimension, which leads to a large overall axial dimension of the tool holder, causing inconvenience for cutting. In order to solve the above problems, it is necessary to design a strain elastic element with compact structure and low cross-sensitivity error for the strain gauge wireless rotating dynamometer to realize online monitoring of the cutting force.

The Maltese cross elastic element is mainly composed of four parts: the outer ring fixed flange, loading platform, cross beam, and floating beam. It has the advantages of compact structure and flexible design structure. On the one hand, scholars have optimized the force measurement performance of the Maltese cross elastic element by changing the structural parameters. The research of Kang [20] and Akbari [21] showed that the thinning of the floating beam thickness and the increase in the cross beam freedom are conducive to improving the structural sensitivity and reducing the cross-sensitivity error, but will reduce the stiffness of the structure. Therefore, some scholars have tried to change the original structure of the Maltese cross elastic element. Kim [22] changed the single-layer floating beam to a double-layer one, which increased the stiffness compared to the single-layer floating beam. However, the cross-sensitivity error reached 2.85% due to the strain gauges for detecting forces in different directions being arranged on the cross beam of the same cross section. Later, Kim [23] distributed axial and tangential square holes on the cross beam, so that different positions of the cross beam have different sensitivities to force in different directions. This design not only improves the sensitivity, but also reduces the cross-sensitivity error to 1.91%. Rizal [24] made a plate dynamometer using the Maltese cross elastic element with square holes distributed axially and tangentially on a cross beam and used it for three-component cutting force measurement, which showed that the Maltese cross elastic element can be used for cutting force measurement.

This paper proposes a compact strain elastic element with double-layer cross floating beam that can be applied to the strain gauge wireless rotating dynamometer, and performs static calibration, dynamic calibration, and cutting tests on the strain elastic element. The results show that the proposed strain elastic element has high sensitivity and low cross-sensitivity error, and can be applied to the strain gauge wireless rotating dynamometer. In addition, this paper provides a certain theoretical basis for the subsequent design and research of the strain gauge wireless rotating dynamometer.

## 2. Design and Research of Strain Elastic Element with Double-Layer Cross Floating Beam

### 2.1. Structural Design of Strain Gauge Wireless Rotating Dynamometer

The overall structure of the strain gauge wireless rotating dynamometer designed in this paper is shown in Figure 1. The overall diameter of the dynamometer is 170 mm and the axial dimension is 286 mm. The structure adopts a modular design, with the tool holder, upper connecting flange, strain elastic element, lower connecting flange, and tool base as the basic structural form. By replacing the connecting flange, different tool holders and tool bases can be connected to suit different processing processes.

For the strain gauge wireless rotating dynamometer, the design and performance of the strain elastic element are the core, as shown in component 4 of Figure 1b. The strain elastic element connects the tool holder and the tool base. During the machining process, the cutting force is transmitted to the strain elastic element through the tool and the tool base. The strain gauge attached to the surface of the elastic element detects the strain under the action of the cutting force, thus obtaining the cutting force during the machining process. The data acquisition and wireless transmission system of the strain gauge wireless rotating dynamometer designed in this paper is shown in Figure 1c, including the signal conditioning module, AD conversion module, main control module, wireless transmission module, and power module. In order to facilitate integration with the wireless rotating dynamometer, the PCB is designed to be ring-shaped.

### 2.2. Structural Design of the Strain Elastic Element

Based on the classic Maltese cross elastic element, a compact strain elastic element with double-layer cross floating beam is designed in this paper. The overall structure is shown in Figure 2a. Figure 2b shows the dimensions of the structure, with a maximum outer diameter of only 75 mm and a maximum thickness of 25 mm. Figure 2c shows the double-layer cross floating beam of the strain elastic element, which is the force-measuring unit of $F_{X(Y)}$ and $F_{Z}$. It improves the stiffness of the overall structure while ensuring sensitivity.

The force measurement principle is as follows: When the loading platform (4) is subjected to $F_{X(Y)}$ and $F_{Z}$, the force is transmitted from the loading platform (4) through the cross beam (1) to the double-layer cross floating beam (2), and finally to the outer ring fixed flange (3). As can be seen from Figure 2c, when the floating beam is subjected to $F_{X(Y)}$, the deformation and strain are mainly generated by Beam 3 and Beam 4. When the floating beam is subjected to $F_{Z}$, the deformation and strain are mainly generated by Beam 1 and Beam 2. It should be noted that the size parameters of Beam 1–4 are the same. When the loading platform (4) is subjected to $M_{Z}$, deformation and strain are mainly generated by the cross beam (1).

### 2.3. Strain and Deformation Segmented Rigid Body Model of Double-Layer Cross Floating Beam

The double-layer cross floating beam is the force measuring unit of $F_{X(Y)}$ and $F_{Z}$ on the strain elastic element. It is not enough to obtain the finite element solution of the strain and deformation of the double-layer cross floating beam under loading by using finite element software, but also to establish a static model. Assuming that the elastic element is ideal, the geometric shape and size of the structure are completely symmetrical on the *X*-axis and *Y*-axis. When the same force ($F_{X}=F_{Y}$) is applied to the elastic element along the *X*-axis or *Y*-axis, the strain and deformation of the elastic element along the *X*-axis and *Y*-axis are the same. Therefore, it is only necessary to analyze the strain and deformation generated by $F_{X}$ (or $F_{Y}$) and $F_{Z}$.

#### 2.3.1. Strain and Deformation Analysis of Double-Layer Cross Floating Beam under the Action of $F_{X}$

When the double-layer cross floating beam is subjected to $F_{X}$ action, its force diagram is shown in Figure 3a. The floating beam unit is divided into two sections, segment AB is regarded as a rigid body, and the section B is regarded as a fixed end. The force diagram of segment BC can be simplified to a single-span statically indeterminate beam as shown in Figure 3b.

The section inertia moment of beam 1–2 is $I_{1-2}=ac^{3}/12$, and the section inertia moment of beam 2–3 is $I_{2-3}=ad^{3}/12$. According to the relevant theories of material mechanics, the bending moment $M_{X}$, axial force $F_{XN}$, and strain distribution $\varepsilon_{X}$ on the upper surface of beam 1–2 at place *x* are, respectively:(1)$$M_{X}(x)=\frac{6\lambda +2}{6\lambda +1}\cdot \frac{F_{X}h}{4}-\frac{F_{X}}{2}x$$
(2)$$F_{XN}=\frac{6\lambda }{6\lambda +1}\cdot \frac{F_{X}h}{2l}$$
(3)$$\varepsilon_{X}(x)=-\left(\frac{z_{1-2}M_{X}(x)}{EI_{1-2}}+\frac{F_{XN}}{EA_{1-2}}\right)=-\left(\frac{z_{1-2}}{EI_{1-2}}\left(\frac{6\lambda +2}{6\lambda +1}\cdot \frac{F_{X}h}{4}-\frac{F_{X}}{2}x\right)+\frac{6\lambda }{6\lambda +1}\cdot \frac{F_{X}h}{2EA_{1-2}l}\right)$$

Among them, $\lambda =I_{2-3}h/I_{1-2}l$, h=e−d/2, l=a−c. $z_{1-2}=c/2$ is the distance from the upper surface to the neutral plane, $A_{1-2}=ac$ is the cross-sectional area of beam 1–2, and E is the elastic modulus of the material.

On the other hand, according to the Timoshenko beam theory, the approximate partial differential equation, angle equation, and deformation equation of the deformation curve of beam 1–2 are as follows:(4)$$w_{X}^{\prime \prime }(x)=-\frac{M_{X}\left(x\right)}{EI_{1-2}}=-\frac{1}{EI_{1-2}}\left(\frac{6\lambda +2}{6\lambda +1}\cdot \frac{F_{X}h}{4}-\frac{F_{X}}{2}x\right)$$
(5)$$\theta_{X}(x)=-\left(\frac{1}{EI_{1-2}}\left(\frac{6\lambda +2}{6\lambda +1}\cdot \frac{F_{X}h}{4}x-\frac{F_{X}}{4}x^{2}\right)+\frac{F_{X}}{2kGA_{1-2}}\right)$$
(6)$$w_{X}(x)=-\left(\frac{1}{EI_{1-2}}\left(\frac{6\lambda +2}{6\lambda +1}\cdot \frac{F_{X}h}{8}x^{2}-\frac{F_{X}}{12}x^{3}\right)+\frac{F_{X}x}{2kGA_{1-2}}\right)$$
where k is the shear coefficient of the beam. For a rectangular cross section, k=10(1+μ)/(12+11μ), where μ is the Poisson’s ratio. G is the shear modulus of the material.

#### 2.3.2. Strain and Deformation Analysis of Double-layer Cross Floating Beam under the Action of $F_{Z}$

When the double-layer cross floating beam is subjected to $F_{Z}$ action, its force diagram is shown in Figure 4a. The overall force diagram can be simplified to a single-span statically indeterminate beam shown in Figure 4b.

Divide the unit into two sections, make segment BC rigid, and calculate the strain and deformation of segment AB. According to the relevant theories of material mechanics, the bending moment $M_{Z}$, axial force $F_{ZN}$, and strain distribution $\varepsilon_{Z}$ on the upper surface of beam 5–6 at place *x* are, respectively:(7)$$M_{Z}(x)=\frac{F_{Z}h}{4}-\frac{F_{Z}}{2}x$$
(8)$$F_{ZN}=\frac{3F_{Z}h}{2l}$$
(9)$$\varepsilon_{Z}(x)=-\left(\frac{z_{5-6}M_{Z}(x)}{EI_{5-6}}+\frac{F_{ZN}}{EA_{5-6}}\right)=-\left(\frac{z_{5-6}}{EI_{5-6}}\left(\frac{F_{Z}h}{4}-\frac{F_{Z}}{2}x\right)+\frac{3F_{Z}h}{2EA_{5-6}l}\right)$$
Among them, $z_{5-6}=c/2$ is the distance from the upper surface to the neutral plane, $A_{5-6}=ac$ is the cross-sectional area of beam 5–6, and $I_{5-6}=ac^{3}/12$ is the section inertia moment of beam 5–6.

On the other hand, according to the Timoshenko beam theory, the approximate partial differential equation, angle equation, and deformation equation of the deformation curve of beam 5–6 are as follows:(10)$$w_{Z}^{\prime \prime }(x)=-\frac{M_{Z}(x)}{EI_{5-6}}=-\frac{1}{EI_{5-6}}\left(\frac{F_{Z}h}{4}-\frac{F_{Z}}{2}x\right)$$
(11)$$\theta_{Z}(x)=-\left(\frac{1}{EI_{5-6}}\left(\frac{F_{Z}h}{4}x-\frac{F_{Z}}{4}x^{2}\right)+\frac{F_{Z}}{2kGA_{5-6}}\right)$$
(12)$$w_{Z}(x)=-\left(\frac{1}{EI_{5-6}}\left(\frac{F_{Z}h}{8}x^{2}-\frac{F_{Z}}{12}x^{3}\right)+\frac{F_{Z}x}{2kGA_{5-6}}\right)$$

When x=h is substituted into Equations (11) and (12), the angle and deformation of section B are obtained as follows:(13)$$\left\{\begin{array}{c} \theta_{B}=-\frac{F_{Z}}{2kGA_{5-6}}\\ w_{B}=-\left(\frac{1}{EI_{5-6}}\cdot \frac{F_{Z}h^{3}}{24}+\frac{F_{Z}h}{2kGA_{5-6}}\right) \end{array}\right.$$

In order to obtain the angle and deformation of section C, segment AB is regarded as a rigid body and segment BC can be simplified as a cantilever beam. The section inertia moment of segment BC is $I_{B-C}$. Then the angle and deformation of the free end of the section BC are as follows:(14)$$\left\{\begin{array}{c} \theta_{B-C}=-\frac{F_{Z}h^{2}}{2EI_{B-C}}\\ w_{B-C}=-\frac{F_{Z}h^{3}}{3EI_{B-C}} \end{array}\right.$$

Then, the angle and deformation of section C can be obtained as follows:(15)$$\left\{\begin{array}{c} \theta_{C}=\theta_{B}+\theta_{B-C}=-\left(\frac{F_{Z}}{2kGA_{5-6}}+\frac{F_{Z}h^{2}}{2EI_{B-C}}\right)\\ w_{C}=w_{B}+\theta_{B}h+w_{B-C}=-\left(\frac{1}{EI_{5-6}}\cdot \frac{F_{Z}h^{3}}{24}+\frac{F_{Z}h}{kGA_{5-6}}+\frac{F_{Z}h^{3}}{3EI_{B-C}}\right) \end{array}\right.$$

### 2.4. Comparison of FEA Solution and Model Solution for Strain and Deformation of Double-Layer Cross Floating Beam

In order to verify the static mechanics model of the double-layer cross floating beam proposed in this paper, the mechanical property parameters of materials in Table 1 and the dimensional parameters in Table 2 were substituted into Equations (3), (6), (9), and (12) and into ANSYS Workbench, respectively, to obtain the model solution and FEA solution. The comparison results are shown in Figure 5.

The following conclusions can be drawn from Figure 5:(1)Ignoring the data mutation caused by the stress concentration at both ends of the strain FEA solution, the strain and deformation model solution are consistent with the FEA solution in the middle section of the beam, indicating that the static model of the double-layer cross floating beam established in this paper is relatively reasonable;(2)By comparing case 1 and case 4, we can see that as *c* increases, the slope of the strain curve on the beam decreases, the extreme value decreases, and the overall deformation also decreases; by comparing case 1 and case 2, we can see that as *d* increases, the slope of the strain curve on the beam basically does not change, but the value shifts downward, and the overall deformation decreases; by comparing case 3 and case 4, we can see that as *a* increases, the slope of the strain curve on the beam decreases, the extreme value decreases, and the overall deformation decreases.

### 2.5. Size Optimization of Double-Layer Cross Floating Beam

The optimization of the size of the double-layer cross floating beam is the key to the design of the strain elastic element, which has great influence on the value and distribution of strain and deformation. The sequential quadratic programming algorithm is selected as the optimization algorithm, and the optimization problem can be expressed as follows:(16)$$t=[t_{1},t_{2},t_{3}]=[a,c,d]$$
(17)$$f(t)=-\frac{\left|\varepsilon_{X}(x_{1})|+|\varepsilon_{Z}(x_{1})\right|}{2}(x_{1}=1.5)$$
(18)$$\begin{array}{c} 8\leq a\leq 10;1\leq c\leq 2;1.3\leq d\leq 2;e=7.5;h=e-d/2;l=a-c;\\ \left|w_{X}(x_{2})\right|\leq 0.006;\left|w_{Z}(x_{2})\right|\leq 0.006;x_{2}=h;\\ \sigma_{MAX}(X)\leq \sigma /3;\sigma_{MAX}(Z)\leq \sigma /3 \end{array}$$

Equation (16) is the independent variable of the objective function, which needs to be optimized in this design. The objective function of the optimization problem is expressed as Equation (17), which is the maximum average value of the strain in the X and Z directions at the position $x_{1}$ = 1.5 mm when the structure is loaded. As can be seen from Equation (18), the value range of independent variables *a*, *c*, and *d* is given, and e = 7.5 mm is determined by the overall structure. Equation (18) also limits the deformation of the double-layer cross floating beam in the X and Z directions to no more than 0.006 mm at the position $x_{2}$ = *h* under loading. The values $\sigma_{MAX}(X)$, $\sigma_{MAX}\left(Z\right)$, respectively, represent the maximum stress of the structure when loaded in the X and Z directions, and σ represents the yield strength of the selected material.

The materials AISI 1045 and $F_{X}=F_{Y}$ = 80 N are selected as an example to optimize the structural parameters. In order to ensure that the global optimal solution is found, six groups of initial design variables are set up for calculation, respectively, as shown in Table 3. The iteration history of each case is shown in Figure 6.

As can be seen from Figure 6, in the results of case 1 to case 6, *c* converges to 1.04 and is rounded to 1 mm; *d* converges to the upper limit of the value range; *a* converges to the lower limit of the value range, and the maximum absolute value of the objective function is 2.6123 × 10^−4^. In order to make the structure suitable for different usage scenarios, it can be obtained by the same logic that when the double-layer cross floating beam is made of structural steel and the force is $F_{X}=F_{Y}$ = 50 N, the optimal parameters of the structure are *a* = 8 mm, *c* = 1 mm, *d* = 1.4 mm, $\left|\begin{array}{c} f(t) \end{array}\right|$ = 2.1588 × 10^−4^. When the material is AISI 5140 and the force is $F_{X}=F_{Y}$ = 80 N, the optimal parameters of the structure are *a* = 8 mm, *c* = 1.6 mm, *d* = 2 mm, $\left|\begin{array}{c} f(t) \end{array}\right|$ = 3.5267 × 10^−4^.

### 2.6. Overall Analysis of the Strain Elastic Element

According to the optimal structural parameters of the double-layer cross floating beam under different materials and loading conditions, the overall structural parameters and range of the strain elastic element under corresponding conditions can be determined. According to the structural characteristics, the range of the strain elastic element is about six to eight times the force of the double-layer cross floating beam. Table 4 shows the optimal structural parameters, range, maximum stress $\sigma_{MAX}$, and safety factor τ of the strain elastic element under different materials. The safety factor is greater than 1.5 to ensure the safety of the structure.

This paper selects the material and force of case 2 for the simulation and physical calibration test. The static analysis of the strain elastic element is performed using ANSYS Workbench. $F_{X}$, $F_{Z}$, and $M_{Z}$ are set to 500 N, 500 N, and 20 Nm, respectively, and the strain cloud diagram of the structure under each force is obtained, as shown in Figure 7.

## 3. Strain Gauge Arrangement and Testing

### 3.1. Strain Gauge Arrangement

According to the finite element strain cloud diagram, the arrangement position of the strain gauge is determined as shown in Figure 8a. 

The strain gauge selected in this paper is BFH-350-1AA-S (Zhuhai Guangce Electronic Technology Co., Ltd. Guangdong, China), with a nominal resistance of 350 Ω. The gauge factor of the strain gauge is 2 ± 1%, the base size is 3.6 × 3.1 mm, and the sensitive grid size is 1.0 × 2.0 mm. In order to avoid the stress concentration area, it should be ensured that the distance between the center of the sensitive grid and the paste edge is about S = 2 mm (Figure 8a).

The strain gauge for detecting $F_{X}$ is SG(1–4), the strain gauge for detecting $F_{Y}$ is SG(5–8), the strain gauge for detecting $F_{Z}$ is SG(9–16), and the strain gauge for detecting $M_{Z}$ is SG(17–20). All strain gauges form four Wheatstone bridges for measurement, as shown in Figure 8b. The relationship between strain *ɛ* and bridge output voltage $U_{o}$ can be expressed as follows [18]:(19)$$\frac{U_{o}}{U_{i}}=\frac{1}{4}GF\varepsilon$$
where *GF* is the gauge factor of the strain gauge and $U_{i}$ is the input voltage of the Wheatstone bridge.

The strain values $\varepsilon_{SGi}$ measured by the strain gauge SGi at each position of the strain elastic element and the strain values $\varepsilon_{F_{X}}$, $\varepsilon_{F_{Y}}$, $\varepsilon_{F_{Z}}$, $\varepsilon_{M_{Z}}$ output by the Wheatstone bridge have the following relationship:(20)$$\left\{\begin{array}{c} \varepsilon_{F_{X}}=\varepsilon_{SG1}-\varepsilon_{SG2}+\varepsilon_{SG3}-\varepsilon_{SG4}\\ \varepsilon_{F_{Y}}=\varepsilon_{SG5}-\varepsilon_{SG6}+\varepsilon_{SG7}-\varepsilon_{SG8}\\ \varepsilon_{F_{Z}}=[(\varepsilon_{SG9}+\varepsilon_{SG11})-(\varepsilon_{SG10}+\varepsilon_{SG12})+(\varepsilon_{SG13}+\varepsilon_{SG15})-(\varepsilon_{SG14}+\varepsilon_{SG16})]/2\\ \varepsilon_{M_{Z}}=\varepsilon_{SG17}-\varepsilon_{SG18}+\varepsilon_{SG19}-\varepsilon_{SG20} \end{array}\right.$$

Figure 9a–c show the strain of the corresponding bridge circuit when the strain elastic element is subjected to $F_{X}$ = 500 N, $F_{Z}$ = 500 N, and $M_{Z}$ = 20 Nm, respectively. This result can provide a reference for subsequent calibration tests and circuit design.

### 3.2. Static Calibration Testing

Within the linear elastic range of the strain elastic element, the strain elastic element can be regarded as a linear system, so the relationship between the applied load $F={\left[\begin{array}{c} F_{X},F_{Y},F_{Z},M_{Z} \end{array}\right]}^{T}$ and the output voltage signal $U={\left[\begin{array}{c} U_{X},U_{Y},U_{Z},U_{M} \end{array}\right]}^{T}$ of the bridge can be expressed by a linear equation:(21)$$\left[\begin{array}{c} U_{X}\\ U_{Y}\\ U_{Z}\\ U_{M} \end{array}\right]=\left[\begin{array}{cccc} S_{11} & S_{12} & S_{13} & S_{14}\\ S_{21} & S_{22} & S_{23} & S_{24}\\ S_{31} & S_{32} & S_{33} & S_{34}\\ S_{41} & S_{42} & S_{43} & S_{44} \end{array}\right].\left[\begin{array}{c} F_{X}\\ F_{Y}\\ F_{Z}\\ M_{Z} \end{array}\right]$$
where matrix ***S*** = {*S_ij_*}(*i*, *j* = 1, 2, 3, 4) is the calibration matrix.

In order to determine the calibration matrix ***S***, a static calibration test is performed using the method shown in Figure 10. Figure 10a shows the Z-direction calibration of the strain elastic element, and the force sensor used is HP-1000 (Yueqing Handpi Instruments Co., Ltd. Zhejiang, China). Figure 10b shows the X, Y, and M_Z_ direction calibration of the strain elastic element, and loading in different directions is achieved by changing the action direction and action point of the two force sensors on the force loading rod. The increase and decrease in the loading force are achieved by the screw and feed nut connected to the force sensor. The force sensor is ZNLBS-V1 (Bengbu Chino Sensor Co., Ltd. Anhui, China). 

In the test, one-dimensional loads are applied to the strain elastic element in four directions ($F_{X}$, $F_{Y}$, $F_{Z}$, $M_{Z}$), respectively, and then the voltage output of the bridge can be obtained in order to obtain the calibration curve as shown in Figure 11.

The calibration matrix is determined by the test as follows:$$\begin{array}{c} S=\left[\begin{array}{cccc} 3.3227 & -0.0077 & -0.0092 & 0.0415\\ -0.0012 & -2.7721 & -0.0175 & -0.0222\\ -0.0075 & 0.0071 & 1.6418 & 0.0253\\ 0.0009 & -0.0062 & -0.0136 & -104.0682 \end{array}\right] \end{array}$$

The cross-sensitivity error is defined as follows [20]:(22)$$C_{ij}=\frac{U_{ij}}{U_{ii}}(i\neq j,i,j=1,2,3,4)$$
where $U_{ij}$ represents the voltage output of the bridge in the *i* direction when the one-dimensional rated load is applied in the *j* direction, and $C_{ij}$ represents the cross-sensitivity error of the one-dimensional rated load in the *j* direction to the bridge output in the *i* direction.

The performances of the strain elastic element obtained according to the calibration tests are shown in Table 5. The calibration results show that the sensitivities of the designed strain elastic element in the $F_{X}$, $F_{Y}$, $F_{Z}$, and $M_{Z}$ directions are 3.3 mV/N, 2.7 mV/N, 1.6 mV/N, and 104.1 mV/Nm, respectively. The maximum cross-sensitivity error is 0.63%, which is lower than 1% and meets the usage requirements [21].

### 3.3. Free Modal Testing

Since machining is a dynamic process, the natural frequency of the strain elastic element must be considered. In order to determine the natural frequency of the elastic element, free mode tests were performed using the natural frequency test system YE6231C (SNOCERA PIEZOTRONICS Co., LTD. Jiangsu, China), as shown in Figure 12a. The accelerometer was fixed on the structure and excited by an impact hammer. 

In the test, the response of the strain elastic element was obtained by the single-point excitation method, and the modal analysis was performed through the signal acquisition system. The test results show that the natural frequency of the strain elastic element is about 2954 Hz, as shown in Figure 12b, which meets the requirements for medium- and low-speed cutting.

### 3.4. Cutting Test

In order to further determine the dynamic performance of the strain elastic element, a cutting test on a fixed platform as shown in Figure 13 was carried out. Test conditions: XK713 CNC milling machine (Wuhan Fourth Machine Tool Factory, Wuhan, China). Tool: 10 mm End Mill. Processing material: 1060 aluminum. Five comparative tests were carried out as shown in Table 6. The test results are shown in Figure 14 and Figure 15.

In Figure 14, by comparing the cutting force and envelope results of case 1, 2, and 3, it can be seen that the cutting force collected by the strain elastic element increases with the increase in cutting depth, indicating that the strain elastic element can obtain the change in cutting force. Figure 15 shows the amplitude–frequency characteristics of the cutting force of cases 4, 5, and 6. The spindle frequency and tool-passing frequency of the three tests are 18.45 Hz, 73.82 Hz, 23 Hz, 92.18 Hz, 27.73 Hz, and 110.8 Hz, respectively. The results show that the strain elastic element can obtain the spindle frequency and tool-passing frequency.

## 4. Conclusions

This paper designed a compact strain elastic element with a double-layer cross floating beam. The maximum outer diameter is only 75 mm and the maximum thickness is 25 mm. The design of the double-layer cross floating beam allows the strain elastic element to improve the sensitivity and reduce the cross-sensitivity error while ensuring the overall stiffness;Based on the proposed strain elastic element, a strain gauge wireless rotating dynamometer with compact size is designed. The overall diameter of the dynamometer is 170 mm and the axial size is 286 mm. In order to facilitate integration, a ring-shaped data acquisition and wireless transmission module PCB is designed;The static model of the double-layer cross floating beam on the strain elastic element was established by the segmented rigid body method. The rationality of the model was verified by comparison with the finite element results. According to the obtained static model, the structural parameters of the double-layer cross floating beam were optimized using the sequential quadratic programming algorithm to maximize the sensitivity of the floating beam;The strain elastic element was analyzed using finite element software, and the strain of the structure under simulation conditions was obtained, which provided a reference for subsequent calibration tests and circuit design;Through static calibration tests, the sensitivities of the strain elastic element in the four directions of $F_{X}$, $F_{Y}$, $F_{Z}$, and $M_{Z}$ are determined to be 3.3 mV/N, 2.7 mV/N, 1.6 mV/N, and 104.1 mV/Nm, respectively, and the maximum cross-sensitivity error does not exceed 1%. Through modal tests in the free state, the natural frequency of the strain elastic element is determined to be 2954 Hz. The results of cutting tests show that the strain elastic element can obtain the change in cutting force and the tool-passing frequency. The results show that the designed strain elastic element can be applied to the strain gauge wireless rotating dynamometer to measure four-component cutting forces under medium- and low-speed conditions.

## Figures and Tables

**Figure 1 micromachines-15-00857-f001:**
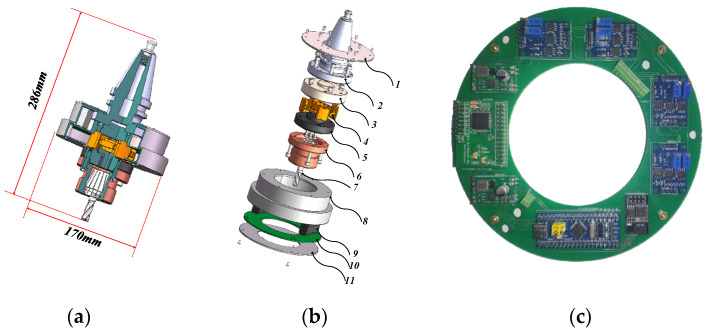
Structure of the strain gauge wireless rotating dynamometer. (**a**) Cross-sectional view of the wireless rotating dynamometer; (**b**) exploded view of the wireless rotating dynamometer; (**c**) data acquisition and wireless transmission system of the wireless rotating dynamometer. 1, Circuit housing connecting plate; 2, BT40 tool holder; 3, upper connecting flange; 4, strain elastic element; 5, lower connecting flange; 6, collet chuck; 7, milling cutter; 8, circuit housing; 9, battery; 10, data acquisition and wireless transmission system; 11, circuit housing cover.

**Figure 2 micromachines-15-00857-f002:**
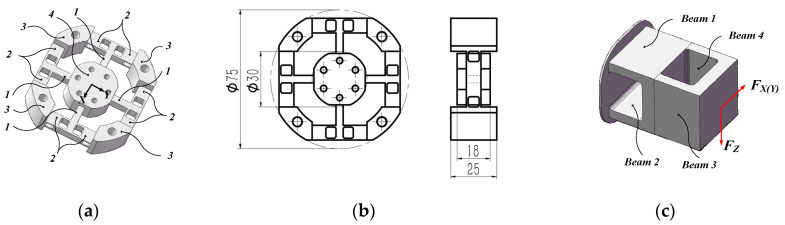
Strain elastic element. (**a**) Strain elastic element; (**b**) structural dimensions; (**c**) double-layer cross floating beam. 1, Cross beam; 2, double-layer cross floating beam; 3, outer ring fixed flange; 4, loading platform.

**Figure 3 micromachines-15-00857-f003:**
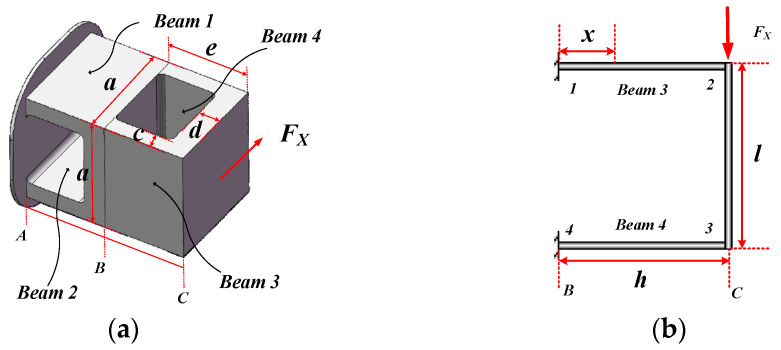
Double-layer cross floating beam is subject to $F_{X}$. (**a**) Double-layer cross floating beam loading $F_{X}$; (**b**) double-layer cross floating beam loading $F_{X}$ simplified.

**Figure 4 micromachines-15-00857-f004:**
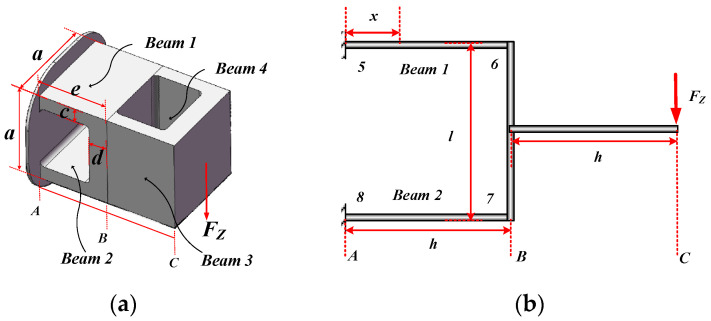
Double-layer cross floating beam is subject to $F_{Z}$. (**a**) Double-layer cross floating beam loading $F_{Z}$; (**b**) double-layer cross floating beam loading $F_{Z}$ simplified.

**Figure 5 micromachines-15-00857-f005:**
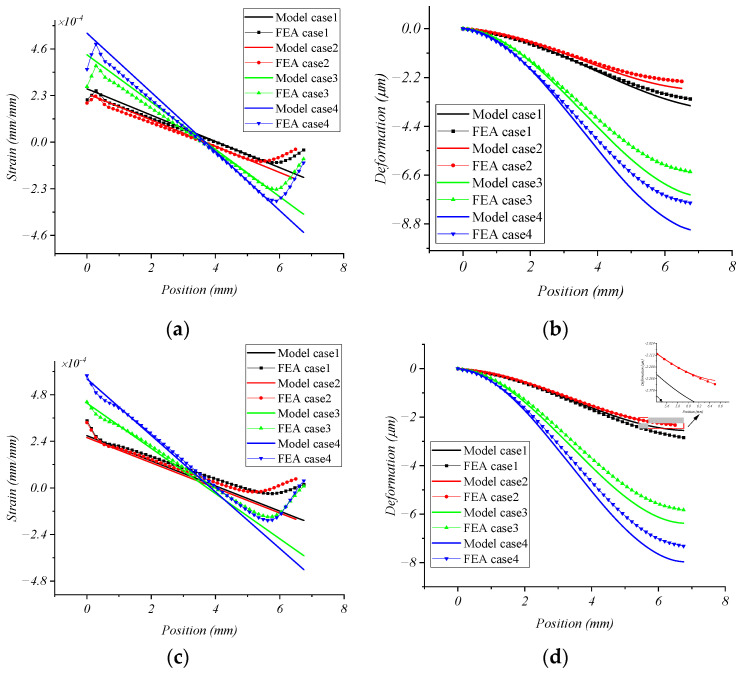
Comparison between FEA solution and model solution for strain and deformation of double-layer cross floating beam. (**a**) Beam strain under $F_{X}$ loading; (**b**) beam deformation under $F_{X}$ loading; (**c**) beam strain under $F_{Z}$ loading; (**d**) beam deformation under $F_{Z}$ loading.

**Figure 6 micromachines-15-00857-f006:**
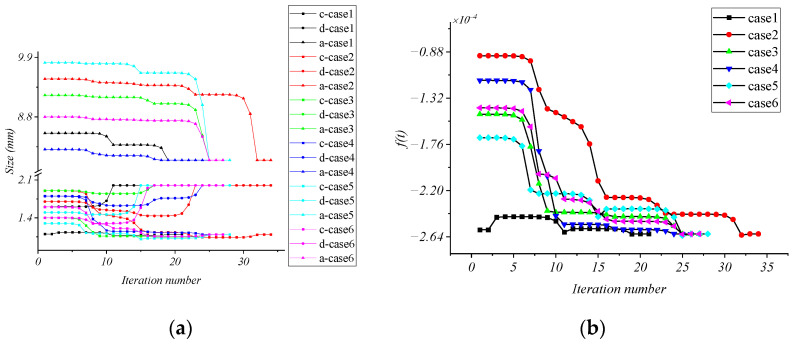
Iteration history of design variables. (**a**) Iteration history of *a*, *c*, *d*; (**b**) iteration history of *f*(*t*).

**Figure 7 micromachines-15-00857-f007:**
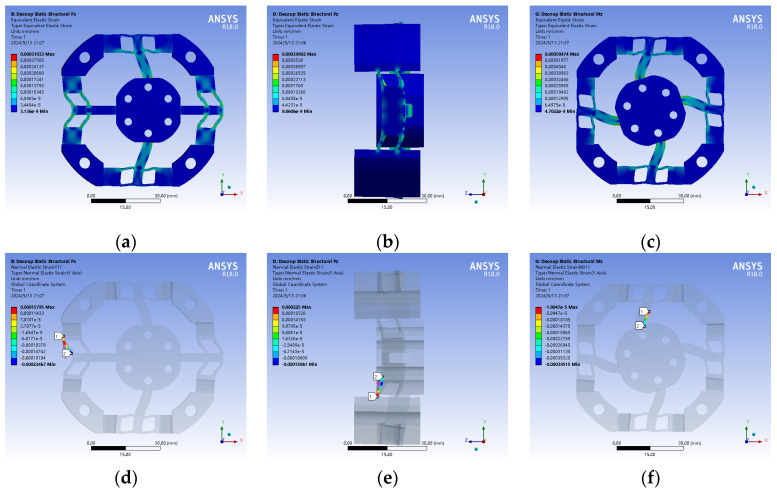
Strain cloud diagram of strain elastic element. (**a**,**d**) $F_{X}$ loading; (**b**,**e**) $F_{Z}$ loading; (**c**,**f**) $M_{Z}$ loading.

**Figure 8 micromachines-15-00857-f008:**
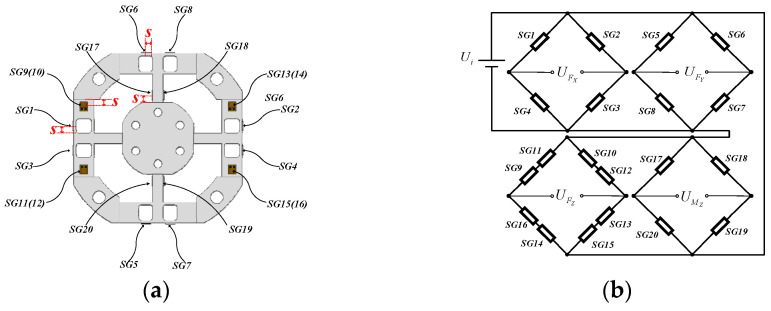
Strain gauge arrangement and Wheatstone bridge of strain elastic element. (**a**) Strain gauge arrangement; (**b**) Wheatstone bridge of $F_{X}$, $F_{Y}$,$\;F_{Z}$, $M_{Z}$.

**Figure 9 micromachines-15-00857-f009:**
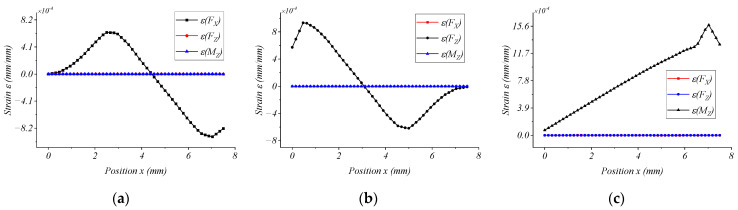
Normal strain of the middle line on the surface of the double-layer cross floating beam of the elastic element. (**a**) $F_{X}$ loading; (**b**) $F_{Z}$ loading; (**c**) $M_{Z}$ loading.

**Figure 10 micromachines-15-00857-f010:**
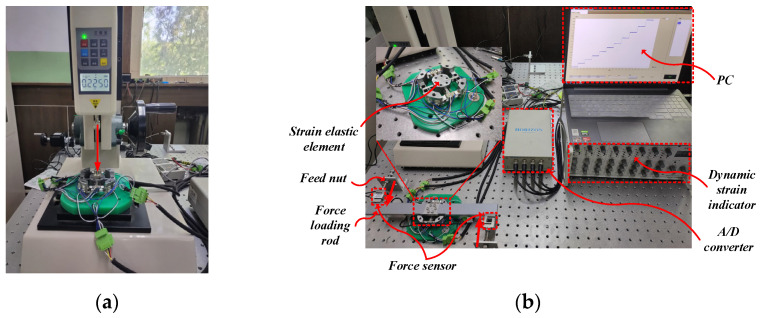
Static calibration test. (**a**) $F_{Z}$ loading; (**b**) $F_{X}$, $F_{Y}$, $M_{Z}$ loading.

**Figure 11 micromachines-15-00857-f011:**
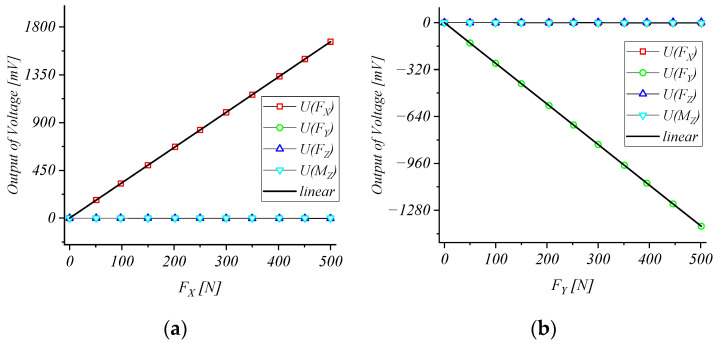
Static calibration test results. (**a**) $F_{X}$; (**b**) $F_{Y}$; (**c**) $F_{Z}$; (**d**) $M_{Z}$.

**Figure 12 micromachines-15-00857-f012:**
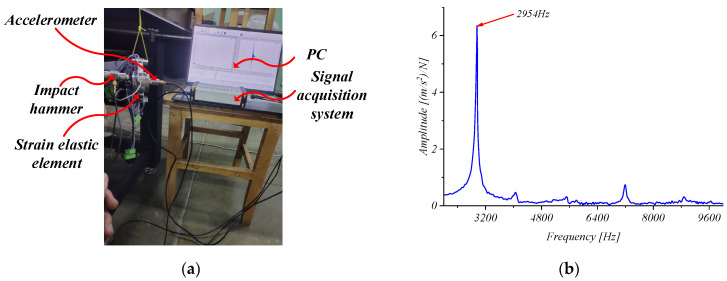
Dynamic calibration test and results. (**a**) Dynamic calibration; (**b**) natural frequency of the strain elastic element.

**Figure 13 micromachines-15-00857-f013:**
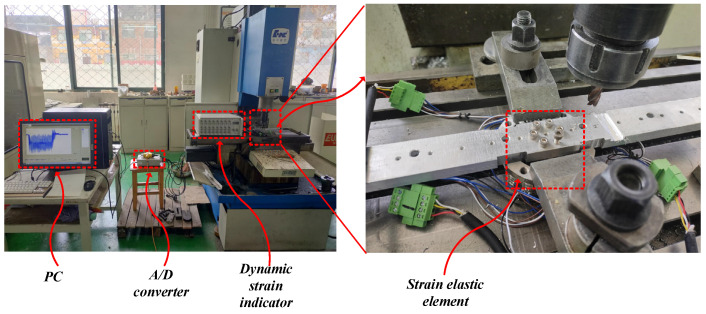
Cutting test.

**Figure 14 micromachines-15-00857-f014:**
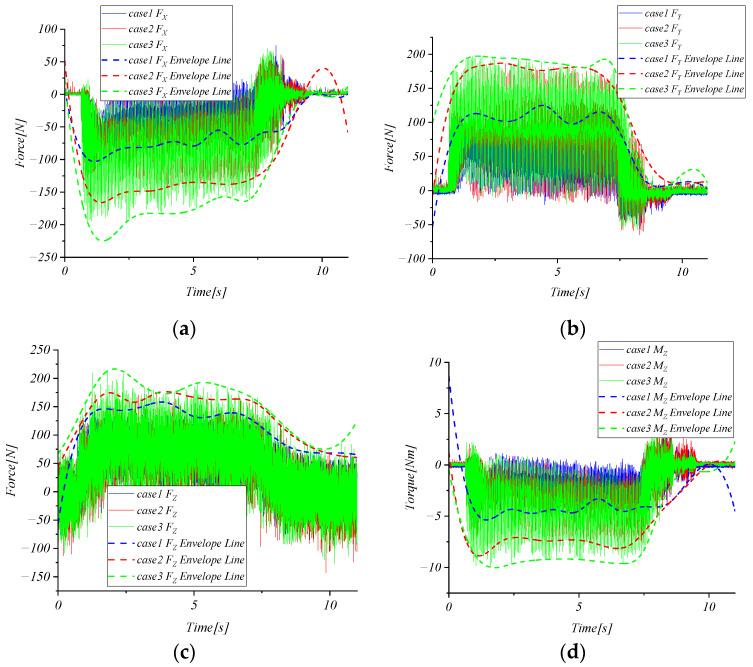
Cutting force curve and envelope line. (**a**) $F_{X}$; (**b**) $F_{Y}$; (**c**) $F_{Z}$; (**d**) $M_{Z}$.

**Figure 15 micromachines-15-00857-f015:**
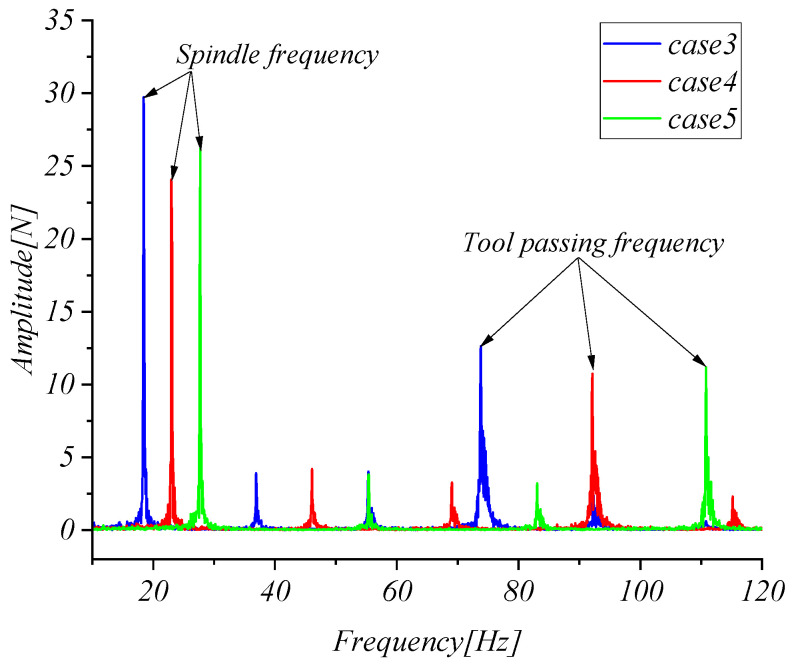
Amplitude–frequency characteristics.

**Table 1 micromachines-15-00857-t001:** AISI 1045 mechanical properties.

Properties	Values
Young’s modulus (GPa) *E*	206
Poisson’s ratio *μ*	0.29
Shear modulus (GPa) *G*	79.8
Yield strength (MPa) *σ*	350

**Table 2 micromachines-15-00857-t002:** Size parameters of the double-layer cross floating beam.

Case	Size (mm)						Force (N)	
	*a*	*c*	*d*	*e*	*h*	*l*	$F_{X}$	$F_{Z}$
case1	8	1.5	1.5	7.5	6.75	6.5	80	80
case2	8	1.5	2	7.5	6.5	6.5		
case3	10	1	1.5	7.5	6.75	9		
case4	8	1	1.5	7.5	6.75	7		

**Table 3 micromachines-15-00857-t003:** Initial values of design variables for double-layer cross floating beam.

Case	Design Variables (mm)				Case	Design Variables (mm)			
	*a*	*c*	*d*	$\left|\begin{array}{c} f(t) \end{array}\right|$		*a*	*c*	*d*	$\left|\begin{array}{c} f(t) \end{array}\right|$
1	8.5	1.1	1.6	2.5735 × 10^−4^	4	8.2	1.8	1.8	1.1519 × 10^−4^
2	9.5	1.9	1.7	0.9169 × 10^−4^	5	9.8	1.3	1.5	1.6964 × 10^−4^
3	9.2	1.4	1.9	1.4732 × 10^−4^	6	8.8	1.6	1.4	1.4128 × 10^−4^

**Table 4 micromachines-15-00857-t004:** Different structural parameters, range, σ, $\sigma_{\mathrm{MAX}}$, and τ of strain elastic element for different materials.

Case	Material	Structural Parameter			Range		Yield Strength (MPa) *σ*	$\sigma_{MAX}$ (MPa)	τ
		*a*	*c*	*d*	$F_{X}$$,F_{Y}$$,F_{Z}$ (N)	$M_{Z}$ (Nm)			
case1	Structural Steel	8	1	1.4	300	10	250	129.82	1.92
case2	AISI 1045	8	1	2	500	20	350	215.9	1.62
case3	AISI 5140	8	1.6	2	1000	40	785	480.9	1.63

**Table 5 micromachines-15-00857-t005:** Sensitivity and cross-sensitivity error of the strain elastic element.

Force Direction	Sensitivity (mV/N)	Cross Sensitivity Error (%)			
		$C_{i1}$	$C_{i2}$	$C_{i3}$	$C_{i4}$
$F_{X}$ (*i* = 1)	3.3	-	0.23	0.28	0.05
$F_{Y}$ (*i* = 2)	2.7	0.04	-	0.63	0.03
$F_{Z}$ (*i* = 3)	1.6	0.45	0.43	-	0.07
$M_{Z}$ (*i* = 4)	104.1 (mV/Nm)	0.02	0.14	0.31	-

**Table 6 micromachines-15-00857-t006:** Cutting test parameter settings.

Case	Spindle Speed (rpm)	Cutting Depth (mm)	Feed Speed (mm/min)
case1	1110	0.2	262
case2	1110	0.4	262
case3	1110	0.6	262
case4	1380	0.6	262
case5	1680	0.6	262

## Data Availability

The data used to support the findings of this study are available from the corresponding author upon request.

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
