# Peer review of "Design and Research of a Strain Elastic Element with a Double-Layer Cross Floating Beam for Strain Gauge Wireless Rotating Dynamometers"

_micromachines, 2024, doi:10.3390/mi15070857_

Round 1

Reviewer 1 Report

Comments and Suggestions for Authors

In this paper, Strain Elastic Element with Double- layer Cross Floating Beam with small size and high sensitivity is proposed, which is of research value for cutting force measurement, and the following is the problem that I have proposed:

1. In the introduction, the authors describe that a double-layer floating beam leads to an increase in stiffness. Is the stiffness of the strain elastic element of the double-layer cross-floating beam strain gauge proposed in this paper taken into account? How is it determined that the stiffness of this structure is currently optimal?

Does the sensor influence the manufacturing accuracy? Please give exact data for the manufacturing process after mounting the sensor?

2. What is the specific method of wireless transmission in the article, Bluetooth? Is there any interference with the signal from fixtures and other metal devices?

3. The maximum value of the horizontal coordinate in Figure 9a-c is between 6 and 8, is this the maximum displacement that can be produced? If convenient, specify the image changes in the text.

4. The heat generated during the cutting process has an effect on the resistive strain gauges, please add experiments related to the effect of temperature on sensors.

Author Response

Thank you very much for taking the time to review this manuscript.

Comments 1: In the introduction, the authors describe that a double-layer floating beam leads to an increase in stiffness. Is the stiffness of the strain elastic element of the double-layer cross-floating beam strain gauge proposed in this paper taken into account? How is it determined that the stiffness of this structure is currently optimal? Does the sensor influence the manufacturing accuracy? Please give exact data for the manufacturing process after mounting the sensor?

Response 1: Thank you for pointing that out. We agree with this comment.

For the double-layer cross floating beam element proposed in this paper, the deformation equation is obtained in the process of establishing the static model. Under the condition of a certain force, the stiffness is inversely proportional to the deformation. In the process of optimization design, we set the maximum strain as the objective function, and the deformation parameter as the limiting condition of the optimal problem to ensure that the maximum strain can be obtained when the structural stiffness meets the requirements.

In response to the rigidity problem you raised in the introduction, we have revised the expression in the introduction. The design of double-layer floating beam increases the stiffness of the structure compared with the design of single-layer floating beam of the same size. The specific changes can be found on page 2, line 65-67. The text before and after change is as follows:

Before: [Kim [22] changed the single-layer floating beam to a double-layer one, which increased the stiffness of the structure.]

After: [Kim [022] changed the single-layer floating beam to a double-layer one, which increased the stiffness compared to the single-layer floating beam.]

Since the strain elastic element has not been fully applied to the wireless rotary dynamometer, the exact data of the manufacturing process can not be given for the time being. However, we have added the test of using strain elastic element for cutting on fixed table to further verify the feasibility of using strain elastic element to measure cutting force. For details, see section 3.4 of the manuscript.

Comments 2:  What is the specific method of wireless transmission in the article, Bluetooth? Is there any interference with the signal from fixtures and other metal devices?

Response 2: Thank you for pointing this out. We agree with this comment.

In the subsequent design of the dynamometer, we will pay special attention to the interference problem of wireless signal transmission. In the past work, the wireless transmission module of the data acquisition system used the ESP8266 chip and used WIFI technology for wireless transmission. In order to ensure that the wireless transmission signal is not affected by the metal electromagnetic shielding, the circuit housing of Part 8 and the circuit housing cover of part 11 in Figure 1b are manufactured using 3D printing technology, and the material is non-metallic material - white resin.

Comments 3:  The maximum value of the horizontal coordinate in Figure 9a-c is between 6 and 8, is this the maximum displacement that can be produced? If convenient, specify the image changes in the text.

Response 3: Thank you for pointing this out. We agree with this comment.

Figure 9 shows the normal strain of the middle line of the surface of a double-layer cross floating beam under rated load. To this end, we have changed the expression. The specific changes can be found on page 10, line 277-278, and the comparison before and after the change is as follows:

Before: [Figure 9a-c show the strain of the corresponding bridge circuit when the strain elastic element is subjected to F_X, F_Z and M_Z respectively.]

After: [Figure 9a-c show the strain of the corresponding bridge circuit when the strain elastic element is subjected to F_X=500N, F_Z=500N and M_Z=20Nm respectively.]

Comments 4: The heat generated during the cutting process has an effect on the resistive strain gauges, please add experiments related to the effect of temperature on sensors.

Response 4: Thank you for pointing that out. This point provides an important idea for our follow-up work. We fully agree with this comment.

Because the strain elastic element has not been fully applied to the wireless rotary dynamometer, the data of the influence of heat in the cutting process on the strain gauge can’t be given for the time being. However, in the subsequent design of the wireless rotary dynamometer, we will carry out work including but not limited to temperature compensation of the strain gauge and integrated temperature measurement of the cutting process.

Reviewer 2 Report

Comments and Suggestions for Authors

It introduces a strain elastic element specially designed to measure the forces generated during the cutting process, featuring this new double-layer cross floating beam. This structure provides high sensitivity and low cross-sensitivity error while ensuring the overall stiffness of the system. The static model of the double-layer cross floating beam is established using the segmented rigid body method and validated using ANSYS finite element software. A sequential quadratic programming algorithm is utilized to optimize the structural parameters, effectively maximizing the sensitivity of the structure.

To enhance the completeness of the paper, the following minor revisions and additional test results are necessary:

  • The graphs in Figures 5, 6, and 9 are too small and lack readability. Increase the size of these graphs to enhance readability.

  • There is no dynamic experimental results during rotation that would validate its applicability in cutting process monitoring systems. Only Free Modal Testing by the impact hammer has been conducted. Add test results conducted by actual cutting operations or simulated vibrations of the real cutting process.

Author Response

Thank you very much for taking the time to review this manuscript.

Comments 1: The graphs in Figures 5, 6, and 9 are too small and lack readability. Increase the size of these graphs to enhance readability.

Response 1: Thank you for pointing this out. We agree with this comment.

We have increased the size of figures 5, 6, and 9 for readability.

Comments 2:  There is no dynamic experimental results during rotation that would validate its applicability in cutting process monitoring systems. Only Free Modal Testing by the impact hammer has been conducted. Add test results conducted by actual cutting operations or simulated vibrations of the real cutting process.

Response 2: Thank you for pointing this out. We agree with this comment.

In this paper, we add the test of using strain elastic element for cutting on fixed table, which further verifies the feasibility of using strain elastic element to measure cutting force. For details, see section 3.4 of the manuscript.

Round 2

Reviewer 1 Report

Comments and Suggestions for Authors

OK,no further comments.